# Predicting and mapping soil available water capacity in Korea

Suk Young Hong[1], Budiman Minasny[2], Kyung Hwa Han[1], Yihyun Kim[1] and Kyungdo Lee[1]

[1] National Academy of Agricultural Science, Rural Development Administration (RDA), Suwon, Gyeonggi-do, Republic of Korea
[2] Department of Environmental Sciences, Faculty of Agriculture and Environment, The University of Sydney, NSW, Australia

## ABSTRACT

The knowledge on the spatial distribution of soil available water capacity at a regional or national extent is essential, as soil water capacity is a component of the water and energy balances in the terrestrial ecosystem. It controls the evapotranspiration rate, and has a major impact on climate. This paper demonstrates a protocol for mapping soil available water capacity in South Korea at a fine scale using data available from surveys. The procedures combined digital soil mapping technology with the available soil map of 1:25,000. We used the modal profile data from the Taxonomical Classification of Korean Soils. The data consist of profile description along with physical and chemical analysis for the modal profiles of the 380 soil series. However not all soil samples have measured bulk density and water content at $-10$ and $-1500$ kPa. Thus they need to be predicted using pedotransfer functions. Furthermore, water content at $-10$ kPa was measured using ground samples. Thus a correction factor is derived to take into account the effect of bulk density. Results showed that Andisols has the highest mean water storage capacity, followed by Entisols and Inceptisols which have loamy texture. The lowest water retention is Entisols which are dominated by sandy materials. Profile available water capacity to a depth of 1 m was calculated and mapped for Korea. The western part of the country shows higher available water capacity than the eastern part which is mountainous and has shallower soils. The highest water storage capacity soils are the Ultisols and Alfisols (mean of 206 and 205 mm, respectively). Validation of the maps showed promising results. The map produced can be used as an indication of soil physical quality of Korean soils.

## INTRODUCTION

There is a global increase in demand for soil data and information for quantitative environmental monitoring and modelling (*Dunne & Willmott, 1996*; *Hartemink, 2008*). Accurate, up-to-date and spatially referenced information on soil properties and processes is essential for within-field, catchment, regional, continental and global food, environmental, and ecosystem management (*Batjes, 1996*; *Dunne & Willmott, 1996*; *Hartemink & McBratney, 2008*; *Holmes et al., 2004*; *Wösten et al., 2013*). Digital soil mapping has

Corresponding author
Budiman Minasny,
budiman.minasny@sydney.edu.au

modernised the way soil properties are mapped (*Malone et al., 2009*; *McBratney et al., 2002*; *Terribile et al., 2011*). This coincides with an enormous leap in technologies that allow accurate measurement and prediction of soil properties (*Mulder et al., 2011*; *Triantafilis, Kerridge & Buchanan, 2009*). Accordingly, there is a global need for making a new digital soil map of the world using state-of-the-art and emerging technologies, which enable estimation of soil properties at a fine resolution through the GlobalSoilMap project (*Sanchez et al., 2009*).

Characterisation of available water capacity (AWC) of soil is important for assessing the soil's physical status and quality. Furthermore, available water capacity is a component of the water and energy balances of terrestrial biosphere (*Milly & Shmakin, 2002*), and is required in ecological studies to establish the response of plant or animal species or communities to environmental conditions (*Basson & Terblanche, 2010*; *Piedallu et al., 2011*). The availability of soil moisture controls the rates of evaporation and transpiration, which in turn can have a major impact on climate. It also controls hydrologic processes such as groundwater recharge, infiltration and overland flow. Soil water holding capacity is one of the most important soil factors for plant growth, influencing carbon allocation, nutrient cycling, and the rate of photosynthesis. Many studies have developed pedotransfer functions (PTFs), which predict soil hydraulic properties such as water retention at field capacity and wilting point from basic soil properties (*McBratney et al., 2002*; *Nemes, Pachepsky & Timlin, 2011*; *Saxton & Rawls, 2006*; *Wösten, Pachepsky & Rawls, 2001*). However fewer studies have mapped available water holding capacity (*Wösten et al., 2013*). At a field extent, *Zheng, Hunt & Running (1996)* used topographic wetness index as a surrogate for profile available water capacity (PAWC) in three areas in the state of Montana, USA. *Leenhardt et al. (1994)* evaluated the effectiveness of soil maps at different scales in predicting PAWC, and found that maps at 1:10,000 and 1:25,000 provided good estimates of soil physical properties. *Cazemier, Lagacherie & Martin-Clouaire (2001)* mapped PAWC in a field in south of France using a combination of two different types of information available in soil databases: imprecise descriptions of soil classes, and pedotransfer functions. At a regional extent, *Malone et al. (2009)* mapped PAWC in the Edgeroi area in New South Wales, Australia using the regression kriging method. Meanwhile, *Poggio et al. (2010)* addressed the issue of uncertainty in the spatial prediction using geostatistical simulation techniques at catchment, regional and national scales in Scotland. At a continental extent, *Kern (1995)* estimated profile PAWC in the USA based on pedotransfer function and soil map. Finally, at a global extent, *Batjes (1996)* produced estimates of global PAWC at a resolution of $0.5° \times 0.5°$ using a global database and soil map of the world. In a separate study, *Dunne & Willmott (1996)* established a global PAWC map based on the FAO soil map. These global maps were used in various global models on climate, terrestrial biosphere, and hydrology (*Knorr & Heimann, 2001*; *Milly & Shmakin, 2002*; *Nijssen et al., 2001*; *Tao et al., 2003*; *Walter, Heimann & Matthews, 2001*).

As recently lamented by *Terribile et al. (2011)*, how can soil data from available soil mapping databases be used effectively for hydrological modelling? This paper will attempt to illustrate the procedures for mapping soil available water capacity in Korea

using available national scale database and maps. In Korea there is a need for detailed information on available water capacity for agricultural and environmental modelling purposes. While digital mapping techniques have been used in Korea to map soil erosion risk (*Jung et al., 2004*; *Jung et al., 2005*; *Park et al., 2011*), and landslide susceptibility (*Lee et al., 2012*), we need an estimate of the water holding capacity for the whole peninsula. The objective of this study is to estimate and map soil available water capacity of Korea using digital soil mapping techniques. In doing so, we will develop pedotransfer functions that predict bulk density, field capacity and wilting point.

## MATERIALS AND METHODS

### Geography and soil data

This study considered the mainland of South Korea, excluding some smaller islands, such as Ulleung, and Dokdo. The area is approximately 100,000 $km^2$, lies between 33.931°N and 38.705°N and 125.436°E and 129.705°E. Korea is located in the humid temperate climatic zone, affected by the influence of both continental and oceanic air masses. National mean annual rainfall is 1,300 mm, ranging from 1000 mm (Daegu area) to 1,800 mm (Jeju area). Approximately half of the annual rainfall occurs during the summer months, June to August, with occasional typhoons. During the summer, although the ambient temperature is high (mean temperature; 20–25°C), and crop canopies are thick, the precipitation exceeds the potential evapotranspiration because of concentrated heavy rainfall. As a result, the base saturation ratios in the majority of soils are rather low. Temperatures in the spring and autumn are mild (mean temperatures of 10–15°C) and winter months are rather cold (monthly mean temperature of −5–2°C), particularly in the central and northern regions. In the winter, spring and autumn, the precipitation is less than in the summer. Despite these seasonal differences, the amount of precipitation and the potential evapotranspiration remain similar year round because of the lower ambient temperatures in the drier seasons.

Korea is a mountainous country. More than two-thirds of the country is occupied by mountains with steep slopes. Plains are subdivided into inland plains, coastal plains and the plains in the narrow valleys. The plains had been under intensive use for agricultural production. The high relief of the land coupled with the heavy downfalls of rain in the summer affects the characteristics of Korean soil very profoundly. Soil erosion has been intense throughout the country for a long time, particularly where the population density is high.

The parent materials of Korean soils are part of the ten recognised geologic systems from different geological time series. Dominant rock types include granitic gneiss (32.4%), granite (22.3%), schist (10.3%), limestone of the Chosun series (Cambrian-Ordovician; 10.1%). The former three geological systems are present in about 60% of the land area and are known as acidic rocks. The fact that the rainfall exceeds the potential evapotranspiration and coupled with the abundance of acidic rocks, results in the wide occurrence of acidic soils in the country. The limestones of the Chosun series are alkaline, thus the soils derived from these rocks tend to be neutral or slightly alkaline. These soils are

**Peer**J

only found near Kangwon Province. Some areas contain sandstone bedrock which results in coarse-textured soils. However, even among the soils derived from the same parent rock, the textures can vary depending upon the location in the soil catena. Soils developed in high elevation areas tend to be coarse due to severe loss of fine particles by erosion, while the soils developed at the locations where soil erosion is not severe tend to be fine-textured. Given, the high variation of soil as function of climate, topography, parent materials and landuse, it is a challenge to produce estimates of soil properties at a fine resolution.

The soil survey was initiated by the Korean Rural Development Agency (RDA), UN, and FAO with a reconnaissance survey, making use of aerial photographs purchased from USA funded by Korea Soil Survey Organization between 1964 and 1967. As a result, soil maps of Korea at scales of 1:250,000 and 1:50,000 were published. Thereafter, RDA adopted the US Soil Taxonomy system and carried out the detailed soil survey between 1968 and 1990 alone. Now, detailed soil maps (1:25,000) are available for entire country in both hard copies and digital format. Furthermore, RDA prepared highly detailed soil maps (1:5,000) between 1995 and 1999 for the entire country. These were digitized and made available for the public through the RDA web site (http://soil.rda.go.kr).

The soil database used in this study was compiled based on the soil profiles in "Taxonomical Classification of Korean Soils" (*NIAST, 2000*), which were mostly collected in the 1970s. It includes soil profile descriptions along with physical and chemical analysis for the modal profiles of the 380 soil series defined in South Korea. Soil chemical and physical properties of each horizon ($n = 1,559$) were recorded, including particle size distribution, moisture retention, organic matter content (OM), cation exchange capacity (CEC), and a limited number of bulk density (BD) data. Soil water retention was recorded for water content at $-10$, $-33$ and $-1500$ kPa on the mass basis ($w$ in g 100 g$^{-1}$) using soils that were ground and sieved to $<2$ mm. The statistics of the data is given in the Table S1. The data show a wide range of distribution for bulk density and water retention. The low bulk density and high water retention soils are due to the Andisols, while the high bulk density and low water retention soils are due to Entisols which have a high sand content ($>70\%$).

A couple of problems arise in using this database to map the AWC:

- Not all soil samples have measured bulk density and $w$ at $-10$ and $-1500$ kPa. Thus they need to be predicted using pedotransfer functions
- Water content ($w$) at $-10$ kPa in the database was measured using disturbed (ground) samples. Thus a correction factor is needed to take into account the effect of bulk density.

## Prediction of bulk density

Only 108 samples contain measured BD, so first we derived a linear model predicting bulk density as a function of organic matter (OM) and sand content. While this model has a reasonable predictive capacity ($R^2 = 0.59$), when we applied this model to the whole datasets, we obtained unreasonable estimates of values between $-2.00$ to $2.50$ g cm$^{-3}$. Therefore we used a more conceptual function as proposed by *Tranter et al. (2007)*

which first estimated the mineral bulk density, $BD_{min}$ as a function of sand content and soil depth. This estimate is then combined with the BD model of *Adams (1973)* which incorporated the effect of organic matter (OM). Andisols which contain a high amount of allophanic materials can have a different response to the model. Therefore we derived a separate model for BD for Andisols using a global dataset (*Tempel, Batjes & van Engelen, 1996*).

## Pedotransfer functions predicting field capacity and wilting point

Available water capacity is defined here as the amount of water held by the soil between field capacity and wilting point. Water content at field capacity is usually measured in laboratory at a potential of $-10$ or $-33$ kPa. Here we use water content at $-10$ kPa to represent field capacity, and at $-1500$ kPa for wilting point. Although field capacity is more of a dynamic property which depends on the soil's hydraulic conductivity (*Romano, Palladino & Chirico, 2011*), because of the unavailability of field measurement, we only used the laboratory data.

In the absence of laboratory measurement, water content at field capacity can be predicted using pedotransfer functions from the soil's particle size distribution, and bulk density or soil structural information (*McBratney et al., 2002*; *Saxton & Rawls, 2006*). Here we developed a linear regression model predicting gravimetric water content at $-10$ kPa ($w_{10}$) and $-1500$ kPa ($w_{1500}$) measured on ground soil samples using basic soil information: clay content, sand content, organic matter content (OM) and cation exchange capacity (CEC):

$$w_{10,1500} = f(\text{sand}, \text{clay}, \text{OM}, \text{CEC}).$$

## Adjustment of field capacity according to bulk density

Water content at field capacity is affected by macroporosity and soil structure (*Sharma & Uehara, 1968*), and therefore measurement is recommended using natural soil clods. Meanwhile water content at wilting point or $-1500$ kPa is not much affected by structure, as most water is held with adsorptive forces, thus it can be measured using disturbed soil samples (*Aina & Periaswamy, 1985*). However in the Korean soil database, water content at $-10$ kPa was measured on ground samples. This is because the samples collected from soil survey were mainly for mapping and classification purposes, and samples for bulk density and soil clods were not collected. This problem is also common in other countries (*Bell & Van Keulen, 1996*). *Bell & Van Keulen (1996)* warned against the use of field capacity data derived from disturbed samples as its measurement overestimates in-situ field capacity for most soils except for coarser textured soils.

Thus water retention data in the Korean soil database which were measured on ground samples need to be adjusted to represent the likely water content at a given bulk density. To build such a model, the only publicly available data that contained such measurements is the soil characterization and profile data from the US National Soil Characterization database (*Soil Survey Staff, 1997*). 301 samples in the database contained water content at $-10$ kPa measured using both natural clods and disturbed samples. From this subset,

274 samples were selected for building the model and the rest (27 samples) were used as validation data. The data are from 141 profiles, and the samples come from A, B and C horizons from various depths in the top 2 m of the profile.

The statistics of the soil properties from the US database is given in Table S2. The database contained measurements of $w_{10\ \text{clod}}$, the percent mass of water retained at suction of 10 kPa, which was measured on natural fabric (clods), and reported on a <2 mm base. $w_{10\ \text{ground}}$ is the gravimetric water content of air dry <2 mm (ground) samples, after equilibration at 10 kPa suction. BD is the bulk density (g cm$^{-3}$) of the <2 mm fraction, with volume being measured after equilibration at $-10$ kPa. We used a linear regression to obtain the estimates of $w_{10\ \text{clod}}$ from $w_{10\ \text{ground}}$ plus other basic soil properties.

## Prediction of profile available water capacity

After adjustment of water retention at $-10$ kPa for bulk density, available water capacity for each layer is calculated as the difference in volumetric water content ($\theta$) between field capacity and wilting point, adjusted by gravel content:

$$\text{AWC (mm)} = (\theta_{10} - \theta_{1500})/100 \times (1 - R_v) \times \text{Thickness of layer (mm)} \quad (1)$$

where volumetric water content ($\theta$ in percent volume) is calculated from gravimetric water content (percent mass) multiplied by bulk density (in g cm$^{-3}$):

$$\theta = w \times \text{BD}. \quad (2)$$

$R_v$ is the volume fraction of gravel (m$^3$ m$^{-3}$), calculated from *Saxton & Rawls (2006)*:

$$R_v = \frac{\alpha R_w}{1 - R_w(1 - \alpha)} \quad (3)$$

where $R_v$ is the mass fraction of gravel (kg kg$^{-1}$), and $a$ is the ratio of soil to gravel density (BD/2.65). The PAWC (profile available water capacity in mm) is calculated as the sum for all the layers up to 100 mm:

$$\text{PAWC} = \sum_{i=1}^{n} \text{AWC}_i. \quad (4)$$

The procedure for estimation of PAWC is summarised in Fig. 1.

## Mapping

The map of the available water capacity to 1 m depth (PAWC) was made by calculating the modal PAWC value for each of the South Korean soil series. The modal values for each soil series were then allocated to each of the map units of the 1:25,000 map. An equal-area spline was utilised to represent the continuous soil depth function of AWC. The equal-area spline function not only fits the water content data with depth, but also disaggregates data obtained from horizon bulk samples into a continuous depth distribution. The spline function consists of a set of local quadratic functions tied together with 'knots'
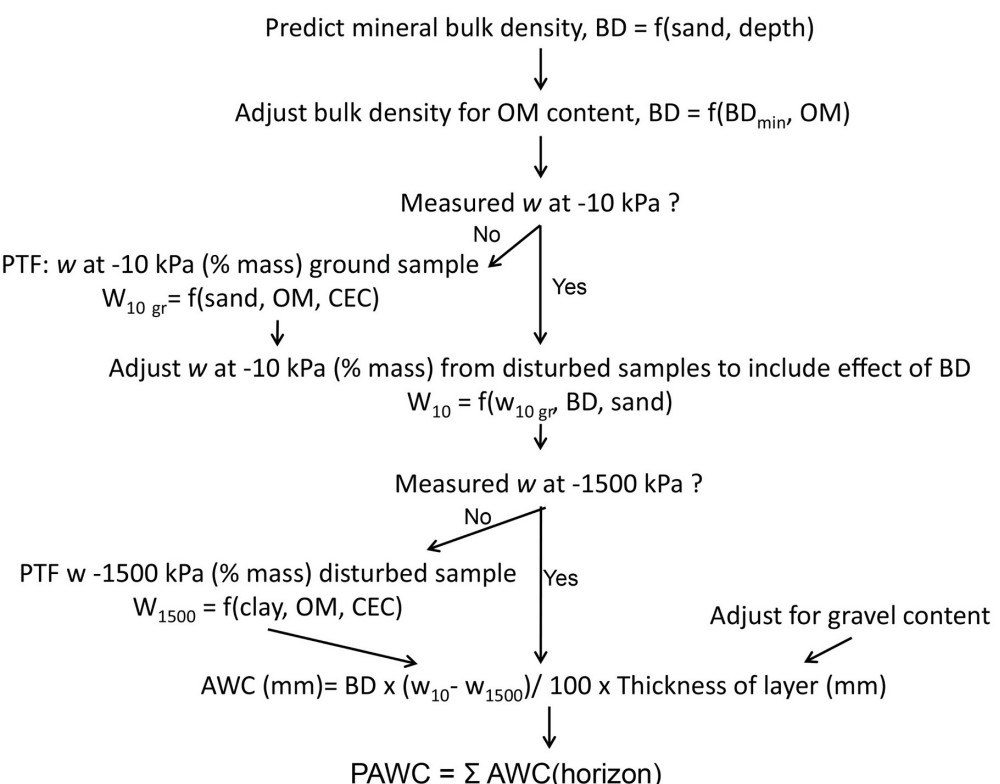

**Figure 1** Procedure for estimation and mapping of profile available water capacity (PAWC) in Korea.

that describe a smooth curve through a set of points. These functions are advantageous for harmonizing soil properties that were collected at different depths. See *Malone et al. (2009)* for the mathematical detail of the functions.

## Map validation

An independent dataset was used to validate the map predictions. The map validation dataset comprises measurement from 170 soil profiles representing 165 soil series. These profiles were collected recently in 2009 and 2011. As the samples were collected for soil survey purposes, no water retention measurement was made, however the data contains measurement of particle size analysis (850 samples) and bulk density (570 samples). We can therefore validate the maps of bulk density, clay and sand content. These variables are the important predictors of available water capacity. The validation was performed on 5 standard depths where the soil samples were measured: 0–5, 5–15, 15–30, 30–60, and 60–100 cm. The prediction from the map was resampled to these standard depths using the equal area spline function.

To validate the prediction of water content at field capacity, another published dataset was used (*Han et al., 2008a*; *Han et al., 2008b*). This dataset, here called water retention validation dataset, is a collection of 18 mostly topsoil (0–20 cm) samples, representing 7 soil series. The samples were measured for soil hydraulic properties using an undisturbed

soil core with the onestep outflow technique, and water content at $-10$ kPa was calculated (see Table S3 for the description of the dataset).

# RESULTS AND DISCUSSION

## PTF for bulk density

Following *Tranter et al. (2007)*, we first derived a pedotransfer function (PTF) predicting mineral bulk density ($BD_{min}$) from sand content and depth based on the limited data in the database ($n = 108$):

$$BD_{min} = 1.017 + 0.0032 * Sand + 0.054 * \log(depth) \tag{5}$$

where depth is the mean depth of the sample (cm) and Sand is the percentage by mass of sand (g 100 $g^{-1}$). The coefficient of determination ($R^2$) is quite low (0.20) as this model only accounts for the influence of the mineral component and overburden. The influence of organic matter on bulk density is then calculated based on the model of *Adams (1973)*:

$$BD = \frac{100}{\frac{OM}{BD_{OM}} + \frac{100 - OM}{BD_{min}}} \tag{6}$$

where OM is the mass percentage of organic matter (g 100 $g^{-1}$), and $BD_{OM}$ = average organic matter bulk density = 0.224 g $cm^{-3}$, and $BD_{min}$ is the mineral bulk density (g $cm^{-3}$).

Because of the different mineralogical composition of Andisols, a separate model that predicted BD from OM content was derived from the *Tempel, Batjes & van Engelen (1996)* dataset:

$$BD = 1.02 - 0.156 \log(OM) \tag{7}$$

($R^2 = 0.45, RMSE = 0.26, n = 642$).

The relationship between OM and bulk density for the soil in the database is given in Fig. 2, which shows bulk density is decreasing with increasing OM content. The relationship for Andisols (Eq. (7)) shows lower bulk density values compared to the other mineral soils. This is to be expected as Andisols are dominated by allophanic materials which have lower density. Applying Eqs. (5), (6) and (7) to the Korean soil database provides an $R^2 = 0.49$ ($n = 108$).

We validated the bulk density PTF using the independent map validation dataset. Table 1 shows the validation statistics in terms of mean error (ME), root mean squared error (RMSE), and coefficient of determination ($R^2$). The results showed that the overall prediction error (RMSE = 0.208 g $cm^{-3}$) is higher compared to values obtained by others. For example *Martin et al. (2009)* obtained an RMSE = 0.123 g $cm^{-3}$ for soils from France and *Tranter et al. (2007)* obtained an RMSE = 0.195 g $cm^{-3}$ for soils in Australia. However, our prediction is not biased (ME values close to 0). The prediction is considered reasonable in this context as the pedotransfer functions were calibrated from a limited observation

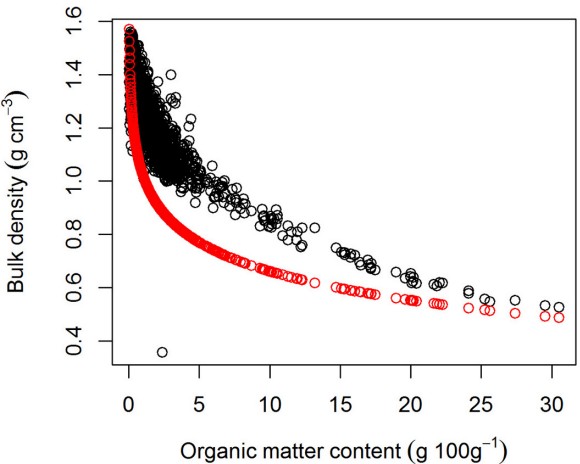

**Figure 2** The relationship between organic matter content and bulk density predicted using pedotransfer functions. The red circles are values predicted for Andisols (Eq. (7)) and the black circles are values predicted for other mineral soils (Eqs. (5) and (6)).

**Table 1** Prediction accuracy for the bulk density PTF tested against an independent dataset.

| Soil | $n$ | ME (g cm$^{-3}$) | RMSE (g cm$^{-3}$) | $R^2$ |
|------|-----|------------------|---------------------|-------|
| All | 426 | −0.0890 | 0.208 | 0.67 |
| Andisols | 56 | −0.0001 | 0.154 | 0.63 |
| Other | 370 | −0.1026 | 0.215 | 0.42 |

**Notes.**

$n$ refers to the number of observations; ME refers to mean error; RMSE is the root mean squared error.

($n = 108$). The prediction for Andisols (Eq. (7)) appeared to be useful with no bias and low RMSE.

## PTF for field capacity and wilting point

From the Korean soil database, we derived a linear model predicting water content at −10 kPa ($w_{10}$) to represent field capacity:

$$w_{10} = 33.18 - 0.188\,\text{Sand} + 0.918\,\text{CEC} + 3.578\,\log(\text{OM}) \tag{8}$$

($R^2 = 0.66$, RMSE $= 8.02$ g 100 g$^{-1}$, $n = 1203$).

Similarly, we derived another model for water content at −1500 kPa ($w_{1500}$) to represent wilting point:

$$w_{1500} = 3.13 + 0.186\,\text{Clay} + 0.541\,\text{CEC} + 1.708\,\log(\text{OM}) \tag{9}$$

($R^2 = 0.61$, RMSE $= 4.67$ g 100 g$^{-1}$, $n = 1435$).

Note that these PTFs are based on measurements from ground samples with units in percentage mass (g 100 g$^{-1}$). The goodness of fit for these PTFs is shown in Supplemental Information. These PTFs are used to fill the data gap of unmeasured $w_{10}$ which are only 25% of the dataset, and unmeasured $w_{1500}$ which constituted 10% of the data.

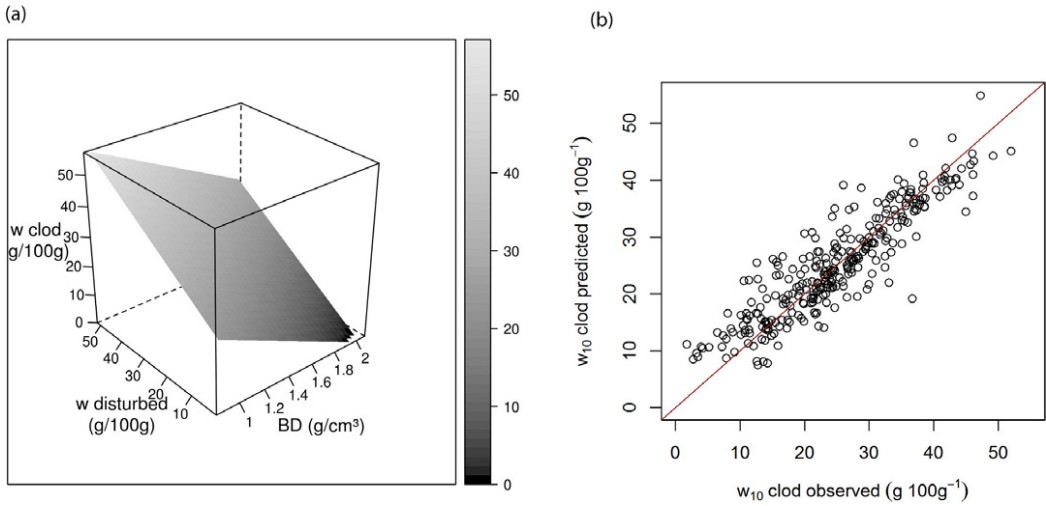

**Figure 3** (A) Response surface of predicted water retention for clods given water retention of ground samples and bulk density, (B) the goodness of fit of the relationship.

## Adjustment of field capacity

Using the dataset from US National Soil Characterization database (*Soil Survey Staff, 1997*), a relationship between $w_{10\,clod}$ and $w_{10\,ground}$ is derived

$$w_{10\,clod} = 40.71 + 0.67\,w_{10\,ground} - 21.36\,BD \tag{10}$$

($R^2 = 0.81$, RMSE $= 4.45$ g $100\,\mathrm{g}^{-1}$, $n = 274$).

This model describes the reduction of soil water content with increasing bulk density (Fig. 3). We validated this model on 27 soil samples in the US database that were not used for deriving the models. For water content at $-10\,\mathrm{kPa}$, Eq. (10) gives an $R^2 = 0.73$. It can be seen that our models predict very well the likely water retention at $-10\,\mathrm{kPa}$ of clod samples given data obtained from disturbed soil samples. This equation allows the estimation of the representative water content at $-10\,\mathrm{kPa}$ for a given bulk density.

## Available water capacity

Based on Eqs. (5) to (10), volumetric water content ($\theta$) at $-10\,\mathrm{kPa}$, $-1500\,\mathrm{kPa}$, and their difference (AWC) were calculated for each soil series. Table 2 shows the predicted values of these soil properties grouped by soil order. Histosols show the highest water retention at $-10$ and $-1500\,\mathrm{kPa}$. Andisols, which are dominated by non-crystalline allophanic minerals, show the highest water storage capacity of 29%, which is also known in the literature (*Shoji et al., 1996*). They also have the highest volumetric water content at field capacity (mean 56%), and the volumetric water content at wilting point (mean 27%). This pattern of high water retention is also observed by *Fontes, Gonçalves & Pereira (2004)* and *Pochet et al. (2007)*. As indicated by *Shoji et al. (1996)*, one of the special characteristics of Andisols is their high water-holding capacity at $-1500\,\mathrm{kPa}$. Entisols and Inceptisols which have loamy texture also show high water retention. The lowest water retention is Entisols which are dominated by sandy materials.

**Table 2** Mean and standard deviation of predicted volumetric water content at field capacity ($\theta - 10$ kPa) and wilting point ($\theta - 1500$ kPa), and predicted available water capacity ($\theta_{10} - \theta_{1500}$), in percent volume, across all soil horizons, grouped by soil order.

| | | $\theta - 10$ kPa | | $\theta - 1500$ kPa | | $\theta_{10} - \theta_{1500}$ | |
| | *n* | Mean | Std. dev. | Mean | Std. dev. | Mean | Std. dev. |
| --- | --- | --- | --- | --- | --- | --- | --- |
| Alfisols | 211 | 36.1 | 8.3 | 14.4 | 5.7 | 21.7 | 7.2 |
| Andisols | 109 | 56.4 | 13.0 | 27.0 | 11.0 | 29.4 | 7.0 |
| Entisols | 184 | 24.7 | 10.1 | 8.1 | 5.4 | 16.6 | 7.2 |
| Histosols | 10 | 57.1 | 19.0 | 30.1 | 7.0 | 27.0 | 12.6 |
| Inceptisols | 946 | 33.5 | 8.6 | 12.5 | 5.8 | 21.0 | 6.9 |
| Mollisols | 6 | 19.3 | 11.6 | 5.1 | 3.1 | 14.2 | 8.5 |
| Ultisols | 133 | 35.9 | 7.3 | 15.2 | 5.1 | 20.7 | 7.3 |

**Table 3** Mean and standard deviation of the available water capacity, in percent volume. Number in brackets represent the number of samples based on the US soil data in *Tempel, Batjes & van Engelen (1996)*.

| Mineralogy[a] | | Coarse textured | Medium textured | Fine textured |
| --- | --- | --- | --- | --- |
| Low activity clay | | 10.5 ± 19.9 (5,274) | 16.2 ± 11.3 (14,541) | 11.3 ± 6.4 (2,663) |
| High activity clay | | 17.2 ± 9.6 (103) | 24.0 ± 28.9 (14) | 18.4 ± 15.1 (50) |
| Mixed activity clay | | 19.0 ± 14.5 (328) | 16.8 ± 13.0 (5,442) | 14.4 ± 7.0 (5,750) |
| Organic | 134.4 ± 737 (184) | | | |
| Allophane | 27.1 ± 14.3 (603) | | | |
| Not specified | 15.0 ± 8.9 (3534) | | | |

**Notes.**
[a] Low activity clay is defined for soils with CEC < 200 mmol $kg^{-1}$, Mixed: between 200 and 620 mmol $kg^{-1}$, High activity > 620 mmol $kg^{-1}$. Organic soils if Organic C content > 16 g 100 $g^{-1}$.

The predicted values were compared with measured AWC values from the US soil data in *Tempel, Batjes & van Engelen (1996)* database. Table 3 shows the AWC of the US soil data grouped by mineralogy and texture. For mineral soils, the highest water capacity is found in allophanic soils with a mean of 27%, similar to our estimates (Table 2). Overall medium textured soils have higher AWC in both soil with low activity and high activity clay. These values are in accordance with our estimates for Korean soils.

Finally, PAWC (amount of available water that can be stored in the soil to 1 m) was calculated using Eq. (1) for the modal soil profiles of the 380 South Korean soil series. Table 4 shows the statistics of the estimates grouped by soil order. Note that the Mollisols and Histosols were excluded in this analysis as they are only represented by 2 soil series. The soils with the highest PAWC are the Ultisols and Alfisols (mean of 206 and 205 mm, respectively). Interestingly, Andisols appear to have a low PAWC, contradictory to the

**Table 4** Mean and standard deviation of profile available water capacity (PAWC) in Korea, grouped by soil order. The average value is based on the area covered by the map.

| | Area (km$^2$) | PAWC (mm) | |
| --- | --- | --- | --- |
| | | Mean | Std. dev. |
| Alfisols | 3,474 | 205 | 27 |
| Andisols | 1,394 | 138 | 54 |
| Entisols | 14,291 | 149 | 73 |
| Inceptisols | 70,868 | 192 | 55 |
| Ultisols | 4,736 | 206 | 55 |

horizon measurement. The low values are due to the higher gravel content of Andisols which affected the calculation of PAWC.

## PAWC map

The value of PAWC for each of the soil series was then mapped onto the 1:25,000 soil map and shown in Fig. 4. The western part of the country generally showed higher available water capacity than the eastern part of the country which is mountainous and has shallower soils. The mean value of PAWC (to 1 m depth) for Korean soil is 189 mm. Total available water capacity is summarized by agricultural land use using a map provided by the Korean Ministry of Land, Transport, and Maritime Affairs. PAWC is the highest in paddy fields, with a mean of 231 mm, which are mostly located in the fluvio-marine plain (See Supplemental Information).

We also are able to display the volumetric water content at $-10$ and $-1500$ kPa down the profile as a continuous function using the equal-area spline. Examples for several soil series are shown in Supplemental Information.

## Map validation

In this paper, each mapping unit (from 1:25,000) is represented by a soil series, and each soil series is represented by a modal profile. Obviously there is variability within a mapping unit, i.e. the soil is not homogenous. Here each soil series is only represented by a modal profile and we do not have an uncertainty measurement of the variability. In order to gauge the accuracy of the map, we compared the map prediction of bulk density, sand and clay content (which were used as inputs for prediction of AWC) with measured values from an independent dataset. Tables 5–7 shows the accuracy measure in terms of mean error (ME), root mean squared error (RMSE), and coefficient of determination ($R^2$).

The map of bulk density (Table 5) is realistic with an overall $R^2$ value of 0.47, and has the same magnitude of error compared to the pedotransfer prediction (Table 1). There is an indication that the prediction at the surface layer (0–5) cm is less accurate, as the surface layer is mostly affected by cultivation. Table 6 shows that the map predicts sand content very well with $R^2$ values between 0.5 and 0.6 for all depths, meanwhile prediction of clay content (Table 7) is less accurate with $R^2$ values between 0.2 and 0.5.

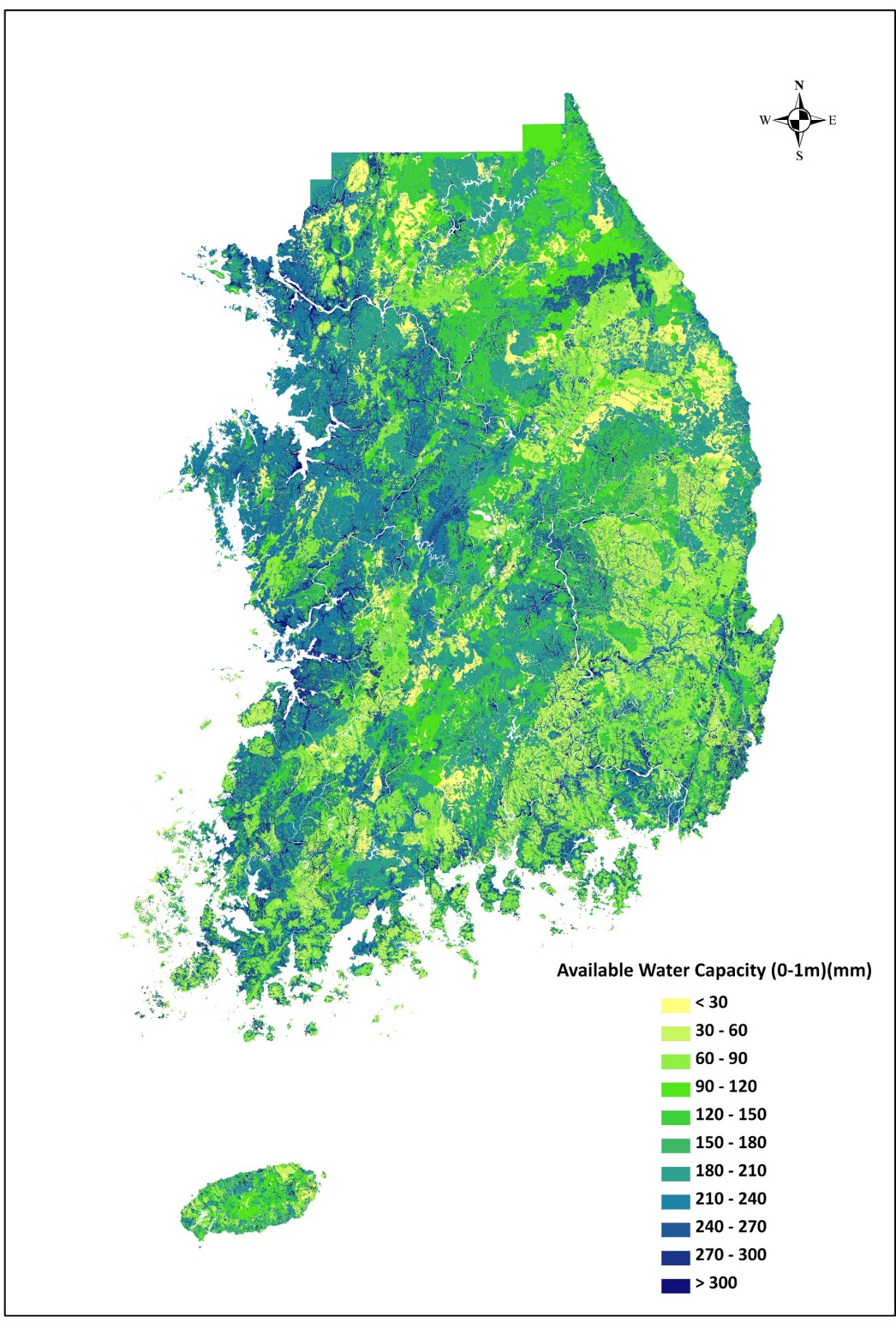

**Figure 4** Map of Korean soil profile available water capacity to 1 m (in mm).

**Table 5** Prediction accuracy for mapped bulk density validated against an independent dataset.

| Depth intervals (cm) | $n$ | ME (g cm$^{-3}$) | RMSE (g cm$^{-3}$) | $R^2$ |
|---|---|---|---|---|
| 0–5 | 111 | 0.013 | 0.225 | 0.36 |
| 5–15 | 111 | −0.002 | 0.221 | 0.41 |
| 15–30 | 113 | −0.057 | 0.252 | 0.47 |
| 30–60 | 111 | −0.059 | 0.260 | 0.44 |
| 60–100 | 98 | −0.057 | 0.238 | 0.43 |
| 0–100 | 566 | −0.032 | 0.239 | 0.47 |

**Notes.**

$n$ refers to the number of observations; ME refers to mean error; RMSE is the root mean squared error; and $R^2$ is the coefficient of determination.

**Table 6** Prediction accuracy for mapped sand content validated against an independent dataset.

| Depth intervals (cm) | $n$ | ME (g 100 g$^{-1}$) | RMSE (g 100 g$^{-1}$) | $R^2$ |
|---|---|---|---|---|
| 0–5 | 167 | 3.26 | 16.1 | 0.53 |
| 5–15 | 167 | 3.62 | 15.6 | 0.55 |
| 15–30 | 167 | 3.46 | 15.3 | 0.57 |
| 30–60 | 163 | 2.60 | 15.3 | 0.60 |
| 60–100 | 167 | 3.26 | 16.1 | 0.53 |
| 0–100 | 831 | 3.50 | 16.0 | 0.56 |

**Notes.**

$n$ refers to the number of observations; ME refers to mean error; RMSE is the root mean squared error; and $R^2$ is the coefficient of determination.

**Table 7** Prediction accuracy for mapped clay content validated against an independent dataset.

| Depth intervals (cm) | $n$ | ME (g 100 g$^{-1}$) | RMSE (g 100 g$^{-1}$) | $R^2$ |
|---|---|---|---|---|
| 0–5 | 167 | −2.00 | 9.05 | 0.24 |
| 5–15 | 167 | −2.06 | 8.53 | 0.29 |
| 15–30 | 167 | −1.73 | 8.46 | 0.40 |
| 30–60 | 163 | −1.71 | 8.46 | 0.52 |
| 60–100 | 150 | −2.06 | 9.79 | 0.46 |
| 0–100 | 814 | −1.90 | 8.86 | 0.40 |

**Notes.**

$n$ refers to the number of observations; ME refers to mean error; RMSE is the root mean squared error; and $R^2$ is the coefficient of determination.

*Leenhardt et al. (1994)* evaluated the accuracy of soil maps in an area of 1328 ha, the south of France. They showed that the $R^2$ values, which were calculated as the proportions of variance explained by the 1: 25,000 soil map, for $w_{10}$, $w_{1500}$ bulk density, clay, and sand content are 0.77, 0.71, 0.73, 0.62, and 0.66, respectively. In comparison, our validation shows lower $R^2$ values for bulk density, clay, and sand content ($R^2$ values of 0.47, 0.56, and 0.40).

**Peer**J

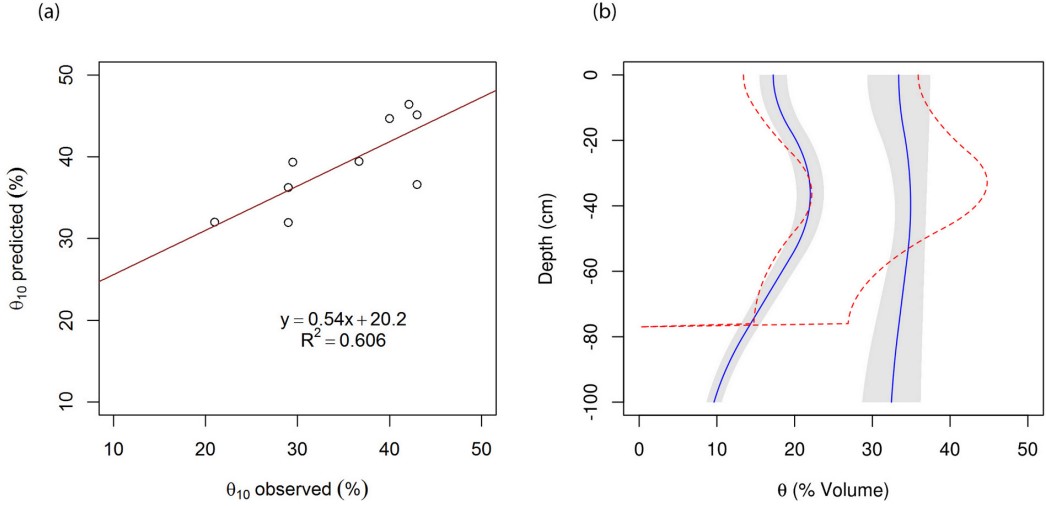

**Figure 5** (A) Measured vs. predicted volumetric water content at −10 kPa on a validation dataset. (B) Measured (blue lines) and predicted (red lines) water retention at −10 and −1500 kPa for a soil profile of Songjeong soil series. The shaded area represents the standard deviation of the water content measurement (of 5 replicates).

Nevertheless, *Henderson et al. (2005)* obtained lower $R^2$ values of 0.44 for the surface soil and 0.22 for sub soil prediction of clay content in Australia. The accuracy of prediction depends on the sampling strategy, landscape condition, extent, and resolution. Nevertheless, the validation of these maps show values that are comparable to other studies which used digital soil mapping techniques employing empirical models that relate soil observations and environmental covariates (*Liu et al., 2012*; *McBratney, Mendonça Santos & Minasny, 2003*).

To test the accuracy of the predicted water content at −10 kPa (field capacity), we compared the prediction of 7 soil series with an independent water retention dataset. Figure 5A shows that the prediction has an $R^2 = 0.61$, and RMSE = 6.3%, which is comparable with values reported by other researchers. *Wösten, Pachepsky & Rawls (2001)* showed that the RMSE values for the prediction of water content at −10 and −33 kPa based on various studies are between 3 and 7%. Because of water retention is rarely measured in Korean surveys, the validation can only be done on a limited soil samples. Nevertheless, Fig. 5B shows an example of the measured and predicted PAWC for a profile from Songjeong soil series. The profile is a fine loamy, mesic family of Typic Hapludults, with measurement collected from soil core samples from Ap1, B1, and C horizons, located in an apple research orchard field at the College of Agriculture and Life Science, Seoul National University, in Suwon. Although there is variation in the depth of the horizons with the modal profile, the prediction of water content at −10 and −1500 kPa fits reasonably well with the measured data.

## CONCLUSIONS

This work demonstrated the derivation of a soil AWC map in Korea using information from conventional soil survey, and soil map integrated in digital mapping procedures.

The process is not straightforward as there are "gaps" between what is available in the soil database and what is required. Thus it required the derivation of pedotransfer functions for bulk density, field capacity and wilting point. Further there is another problem on the use of ground soil samples in measuring field capacity. These substantial data gaps required the derivation of pedotransfer functions from different data sources. The price to be paid for these knowledge gaps is uncertainty in the predictions. The uncertainty in the PTF models mostly comes from uncertainty in estimating BD, followed by estimating water content at $-10$ kPa from ground samples. The other variables have the lowest uncertainty (PTFs for $w_{10}$ and $w_{1500}$) as demonstrated by the higher accuracy of prediction and they are only used in 10–25% of the data.

Validation of the 1:25,000 soil map showed that bulk density, clay, and sand content maps show that the maps can capture 40 to 55% of the soil variability, which is what is to be expected on digital soil maps. We haven't been able to fully validate the AWC map or derive estimates of uncertainty for the map predictions. Uncertainty of the map has several components: uncertainty in the soil map, the representativeness of the modal soil profile, and finally the accuracy of the pedotransfer functions. All these need to be validated, however we do not have enough resources to collect new samples and analyse the water retention relationships. Because of the measurement of water retention on undisturbed cores is time consuming and expensive, it is rarely gauged in Korean survey. The only available data came from a limited number of soil physics experiments. Nevertheless an examination of the predictability of water content at $-10$ kPa on a limited soil series, indicated a reasonable prediction ($R^2 = 0.60$).

We used water content at $-10$ kPa as the upper limit of available water or field capacity. In reality, field capacity is more of a process-dependent parameter, i.e. it depends on soil-water flow (*Romano, Palladino & Chirico, 2011*). Future work needs to measure the field capacity values in the field in the dominant soil types of the study area. There are also problems dealing with Andisols. They can desiccate if dried from 1 kPa to 1500 kPa which leads to significant shrinkage (*Dorel et al., 2000*).

This map gives an indication of PAWC based on the South Korean soil series and a detailed soil map. It will enable us to perform functional evaluation such as running simulation models to predict potential evapotranspiration. This map should provide more detailed and accurate AWC data for hydrological models than global maps of AWC such as the one produced by *Dunne & Willmott (1996)* or *Batjes (1996)* that were based on FAO soil classes.

## ACKNOWLEDGEMENTS

The authors thank Dr. Nathan Odgers for his suggestions on the draft manuscript.

### Funding

This study was supported by PJ00936703, Rural Development Administration, Republic of Korea. Budiman Minasny is supported by the Australian Research Council Discovery

Project Methodologies for global soil mapping. The funders had no role in study design, data collection and analysis, decision to publish, or preparation of the manuscript.

## Grant Disclosures

The following grant information was disclosed by the authors:

PJ00936703, Rural Development Administration, Republic of Korea.

Australian Research Council Discovery Project Devising a methodology for digital soil map of the world.

## Competing Interests

Budiman Minasny is an Academic Editor for PeerJ.

## Author Contributions

- Suk Young Hong conceived and designed the experiments, performed the experiments, analyzed the data, contributed reagents/materials/analysis tools, wrote the paper.
- Budiman Minasny conceived and designed the experiments, analyzed the data, wrote the paper.
- Kyung Hwa Han, Yihyun Kim and Kyungdo Lee performed the experiments, contributed reagents/materials/analysis tools.

## Supplemental Information

Supplemental information for this article can be found online at http://dx.doi.org/10.7717/peerj.71.

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
