# Peer review of "Predicting and mapping soil available water capacity in Korea"

_PeerJ, doi:10.7717/peerj.71_

## Round 0.1 · original submission · Major Revisions

· Academic Editor

Major Revisions

Dear Authors,

This is a potentially interesting paper, however, some serious revisions are required. Firstly, pay due attention to explaining and estimating the degree of uncertainty of presented results, this is really essential to better value the paper output. Secondly, I would encourage the authors to include some new additional material in the paper to validate obtained results through comparing these with acquired field measurement data, at least for some of the major Korean soil types. Thirdly, as indicated by the two reviewers, include only the really essential Figures and Tables and delete all others accordingly.

Reviewer 1 ·

Basic reporting

Please use line numbering.

Page 2: please check the language(e.g. ‘was’ used, where ‘were’ is meant, ‘developed of pedotransfer’ delete the ‘of’ etc.).

Page 10: Here, you apparently mixed the numbering of tables, please check this.

Page 11: The same as previous, but here is something wrong with the Figure numbering. Page 12, here as well some mixing of table numbers, 5 is meant where 4 is written.

Experimental design

The authors present a methodology to predict, based on limited and partly inadequate data, available soil water content. Pedo-transfer functions are used to fill the gaps, and a map showing the AWC is presented. The approach of the study is feasible, but the authors should indicate the uncertainty of the methodology better.

Validity of the findings

In general ok, but see remarks in 'general comments to author' section

Page 2: about 15 lines from bottom: in what way has soil water retention a major impact on climate? Please change this, since Dunne and Willmott (1996) do not state this in their article, they state that ‘…soil moisture … is the main source of water for plant development and terrestrial evapotranspiration. Evapotranspiration and soil moisture, in turn, are sensitive to the potential amount of water the soil can hold and make available to the atmosphere’.

Page 6, last part: Romano et al did NOT state that field capacity depends on its water conductivity. In fact, if you think it is you should explain this, since this sounds strange to me.

Page 6: you did not define your AWC, but apparently you use the water content between -10 and -1500 kPa. Why not from -33 - -1500 kPa, as is suggested in Romona et al, 2011, being the average between the two estimated extremes (-10 for coarse and -50 for fine medium).


Page 14, in the conclusion you write in the end about uncertainty. This part belongs to the discussion, and should have more focus. I realize that limited resources cause lack of possibility to quantify the uncertainty, but you should be able to indicate al sources that affect the uncertainty in your procedure. So, not only uncertainty in the map, but also in your methodology of using PTF’s at several stages.

Additional comments

Specific remarks:

Page 3: second paragraph: please indicate why you develop PTF’s yourself, and don’t use existing functions.

Page 3. Include a figure with location of Korea.

Page 12: Can you indicate the consequences of your remark about Andisols (last sentence, third paragraph)?

·

Basic reporting

• The paper is well structured and pleasant to read, however, it could be improved with the help of an English corrector.

• On page 10 Table 1 is mentioned, presumably this must be Table 3 as this table shows the statistics.

Experimental design

Overall the paper presents a nice conceptual framework as also illustrated in Figure 1, how to proceed from an existing soil map to a map showing available water. However, in the case of Korea substantial data gaps do exist. At present these gaps are filled with pedotransfer functions which often have been derived from different data sources. The price to be paid for these knowledge gaps is probably a considerable uncertainty in the predictions made.

Validity of the findings

• The biggest problem with the paper is that is couples a number of uncertainties by using PTFs with as final result conclusions that could be very uncertain. For instance, equations 5 and 6 predict bulk densities for non-Korean soils. Next, the bulk densities are used in equation 9 to adjust the field capacity measured on grounded soil material to field capacity for structured soils. In doing this a relationship derived for American soils is used. Next field capacities and wilting points are used to predict soil available water for Korean soils. In this whole process is should be made clear how the uncertainties in the separate PTF steps are influencing the uncertainty in the final estimate of soil available water.

• The PAWC map for Korea is not validated because no measured data exist to do so. The most that is achieved is that AWC values calculated for Korean soils are compared to similar data for American soils. Based on this some general trends can be compared (page 12). As a result, it would be good to state that the map provides some indicative estimates which best can be presented in classes as is also done in Figure 5 of the map.

• Given all uncertainties presenting mean PAWC values as is done in Tables 7 and 8 is not encouraged.

• Validation of the map refers really to a validation of the soil map and not to the PAWC map. This could be stated more clearly.

---

## Round 0.2 · Minor Revisions

· Academic Editor

Minor Revisions

The current version of the paper is almost acceptable for publication, and the reviewers comments have been addressed adequately. The only points which need to be clarified and resolved are:

* The missing Figure Captions.
* With this regard, it is also unclear what the line is representing in Figure 2. Is that the relationship between bulk density and organic matter content for Andisols, derived on basis of the data of Tempel et al (1996), and represented by equation 7 in the manuscript?
* In the paper Figures 6 and 7 are not referred to. Either delete these figures, or clearly discuss these figures in the paper.

These relatively minor points need to be addressed before the paper can be published.

---

## Round 0.3 · accepted · Accept

· Academic Editor

Accept

I am pleased that all remarks and suggestions of the reviewers have been addressed accordingly by the authors, and trust that the final version of the paper will be of interest to a wide.range of scientists involved in soil and water related disciplines.